# Potential Associations between Vascular Biology and Hodgkin’s Lymphoma: An Overview

**DOI:** 10.3390/cancers15215299

**Published:** 2023-11-06

**Authors:** Wellington Francisco Rodrigues, Camila Botelho Miguel, Melissa Carvalho Martins de Abreu, Jamil Miguel Neto, Carlo José Freire Oliveira

**Affiliations:** 1Postgraduate Course in Tropical Medicine and Infectious Diseases, Federal University of Triangulo Mineiro, UFTM, Uberaba 38025-440, MG, Brazil; camilabmiguel@hotmail.com (C.B.M.); carlo.oliveira@uftm.edu.br (C.J.F.O.); 2University Center of Mineiros, Unifimes, Mineiros 75833-130, GO, Brazil; dramelissa@unifimes.edu.br (M.C.M.d.A.); jamil@unifimes.edu.br (J.M.N.)

**Keywords:** Hodgkin’s disease, cell biology, blood vessels, targeted therapy, animal models

## Abstract

**Simple Summary:**

Hodgkin’s lymphoma is a type of cancer that affects the lymphatic system, which is an important part of the body’s immune system. This type of cancer involves a set of molecules and cells that promote tumor survival. Some structures found in the tumor niche are associated with vascular biology, which integrates a network of molecules and cells that play a fundamental role in nutrition and oxygen supply to tissues, including tumors. Thus, the main objective of this study was to improve our knowledge of the relationship between vascular biology and Hodgkin’s lymphoma and to enable new studies on disease intervention. Here, we describe the main cells and molecules associated with tumors that play a role in vascular biology. Additionally, this study addresses the main models of disease experimentation.

**Abstract:**

Hodgkin’s lymphoma (HL) is a lymphatic neoplasm typically found in the cervical lymph nodes. The disease is multifactorial, and in recent years, the relationships between various vascular molecules have been explored in the field of vascular biology. The connection between vascular biology and HL is intricate and the roles of several pathways remain unclear. This review summarizes the cellular and molecular relationships between vascular biology and HL. Proteins associated with various functions in vascular biology, including cytokines (TNF-α, IL-1, IL-13, and IL-21), chemokines (CXCL10, CXCL12, and CCL21), adhesion molecules (ELAM-1/VCAM-1), and growth factors (BDNF/NT-3, platelet-derived growth factor receptor-α), have been linked to tumor activity. Notable tumor activities include the induction of paracrine activation of NF-kB-dependent pathways, upregulation of adhesion molecule regulation, genome amplification, and effective loss of antigen presentation mediated by MHC-II. Preclinical study models, primarily those using cell culture, have been optimized for HL. Animal models, particularly mice, are also used as alternatives to complex biological systems, with studies primarily focusing on the physiopathogenic evaluation of the disease. These biomolecules warrant further study because they may shed light on obscure pathways and serve as targets for prevention and/or treatment interventions.

## 1. Introduction

Hodgkin’s lymphoma (HL) is a lymphatic neoplasm typically found in the cervical lymph nodes. HL can be categorized into two types: classic and predominantly lymphocytic nodular HL [1]. According to 2020 data, there were 83,087 new HL cases and 23,376 HL-associated deaths worldwide. It accounts for 0.4% of all cancer-related diagnoses and 0.2% of cancer-related deaths [2]. In the same year, the incidence and mortality rates of HL were 0.98 and 0.26 per 100,000 individuals, respectively [3].

The mortality rate of HL is associated with the economy of the country and the prevalence of smoking, obesity, and hypertension. Specifically, the mortality rate was higher in low-income countries than in high-income countries. Conversely, the incidence of HL was higher in high-income countries. An increasing trend in the incidence of HL has been observed, influenced by factors such as female sex, younger age groups, and Asian countries [3].

The disease is multifactorial and involves numerous molecules and signaling pathways involved in tumor development and resistance to treatment [4,5]. Although further studies are required on certain HL pathways, existing data suggest that factors associated with genetic abnormalities, immunosuppression, and exposure to risk factors, including ionizing radiation, carcinogenic chemicals, and oncogenic viruses, contribute to disease development [5].

Several genetic loci associated with the susceptibility to HL have been linked to HL pathogenesis. Exome sequencing has shown mutations in genes linked to the disease, including HLA, B2M, NFKBIA, STAT3, and STAT6 [6,7,8]. Infections with Epstein–Barr virus (EBV) are associated with a greater risk of developing Hodgkin’s lymphoma, although questions remain regarding the association between viral infection and the pathogenesis of classic HL [9,10]. There is evidence that EBV is linked to different pathogenic effects of the disease at different stages of life [10] and cells infected by the EBV express LMP1, which also activates and enhances the NF-κB pathway [11].

Several mechanisms contribute to the pathogenesis of classical HL, including the altered expression of B-cell markers, which are virtually absent in classical HL, possibly owing to the downregulation and epigenetic inactivation of B-cell genetic regulators and control factors. Furthermore, the NF-κB, JAK-STAT, and PI3K-AKT-mTOR signaling pathways were hyperactive [11]. NF-κB activity is promoted by somatic mutations, amplifications of REL, MAP3K, and BCL3, and the inactivation of the negative regulators NFKBIA, NFKBIE, and TNFAIP3. The JAK-STAT pathway is activated by JAK and STAT mutations via inactivation of PTPN1 and SOCS1 and activation of the interleukin receptor. Cells in the tumor microenvironment also interact with tumor cells via cytokines and chemokines. Mutations in genes, such as CD274 (encoding PD-L1), help tumor cells evade the immune response [11].

The onset, maintenance, and survival of patients with HL are contingent on a favorable microenvironment resulting from various protein structures associated with vascular homeostasis [12]. Vascular biology related to tumors has significant relevance for tumor growth and preservation as well as for containment and intervention. However, numerous gaps remain regarding this relationship, necessitating more comprehensive evaluations [13,14]. These evaluations should be conducted through secondary studies that facilitate the summarization and generation of consistent indicators from the available literature, as well as through new clinical studies. A detailed understanding of the signaling pathways involved in HL could shed light on disease pathogenesis and generate indicators for prevention and drug intervention [15,16].

The intricate relationship between vascular biology and HL involves numerous pathways that remain unclear. A more comprehensive understanding of this association can be achieved by examining the roles of various molecules within the tumor microenvironment and vascular system. These include endothelial factors, platelets, coagulation cascade proteins, the fibrinolytic pathway, lymphoid and myeloid lineage cells, and various molecules associated with signaling pathways. These elements collectively contribute to the interplay between vascular biology and HL [17,18,19].

In this review, we examined and summarized the cellular and molecular associations between vascular biology and HL. Furthermore, we addressed the primary features of preclinical biological models of HL. Information from the last few decades was collected from the Medline/PubMed database related to terms and synonyms for vascular biology and HL.

## 2. Brief History of HL

The study of HL has a rich and progressive history spanning several decades. Recent years have witnessed significant strides in our understanding of its pathophysiology, largely owing to technological advancements in science. These developments have also led to improvements in intervention protocols [20].

HL derives its name from the British physician Thomas Hodgkin, who meticulously described the disease in 1832 [21]. His description included the pathological findings of patients exhibiting enlarged lymph nodes, notably the presence of multinucleated giant cells [21].

Throughout the 20th century, advancements in pathology and microscopy have facilitated the improved characterization of HL [22]. Despite these advancements, the true nature of Reed cells, a defining feature of the disease, has remained elusive for many years. The 1960s saw the advent of immunofluorescence and immunohistochemical techniques, which enabled the accurate identification of Reed–Sternberg cells [23]. These cells, characterized by their multinucleated lacunar structures, are the hallmarks of HL. Over time, these have been acknowledged as the key features of this disease [22].

Advances in molecular biology during the 1980s and 1990s have significantly enhanced our understanding of the genetic alterations associated with HL [24,25]. The discovery of chromosomal abnormalities and genetic mutations in Reed–Sternberg cells has provided a more comprehensive understanding of disease pathogenesis [26].

Between the 1980s and 1990s, substantial progress was made in treatment methodologies, including the successful application of combined chemotherapy and radiotherapy [27]. Integration of these strategies has enhanced the survival rates of numerous patients diagnosed with HL.

Research in the 21st century has elucidated the molecular and immunological aspects of HL. The understanding of signaling pathways implicated in the proliferation, survival, and interactions with the microenvironment of Reed–Sternberg cells has been enhanced [28]. Concurrently, personalized therapeutic strategies, including immunotherapy and targeted therapies, have been developed. Immunotherapy utilizing monoclonal antibodies, such as brentuximab vedotin, specifically target Reed–Sternberg cells. In contrast, targeted therapies target specific signaling pathways implicated in the disease [29].

Research on HL is continually advancing, with a growing emphasis on the identification of prognostic and predictive biomarkers and the comprehension of treatment resistance mechanisms. New therapies and strategies, aimed at enhancing patient treatment and quality of life, are being developed [30,31,32].

In conclusion, the history of HL research has demonstrated consistent progress in discoveries, technological enhancements, and therapeutic advancements. These studies have deepened our understanding of the disease and have improved treatment options over time. To enhance our understanding of the evolution of HL studies, we conducted a survey of the Medline/PubMed database spanning 1940–2021. We used the Hodgkin disease descriptor and all registered synonyms in the Medical Subject Headings. A total of 214,829 studies were identified, yielding an annual average of 2620 indexed studies in the evaluated databases, ranging from a minimum of one to a maximum of 10,524 (Figure 1).

On evaluating the annual distribution, a positive and significant temporal correlation was observed across all assessed decades and the total distribution (Figure 1). The most pronounced increase began in the 1960s, followed by a substantial increase after the 2000s. The data gathered regarding escalation in studies likely correlates with advancements in science and technology since the 1980s [33,34].

## 3. General Characteristics of HL

The appearance of Hodgkin cells is a classic characteristic of HLs. When multinucleated, these cells are referred to as Reed–Sternberg cells. The typical Reed–Sternberg cell is a large entity, potentially exceeding 50 μm in diameter. These cells are binucleated, featuring prominent eosinophilic nuclei encased in an abundant cytoplasm [35,36,37].

The enhanced understanding of Reed–Sternberg cells was pivotal in establishing their direct association with HL, a connection initially suggested by pathologists Dorothy Reed and Carl Sternberg [35,38]. Reed–Sternberg cells derived from B lymphocytes possess unique characteristics and genetic alterations that distinguish them from conventional B lymphocytes [35,39]. While both Reed–Sternberg cells and B lymphocytes are found in the lymphatic system, Reed–Sternberg cells are exclusive to malignant tumor cells and exhibit distinctive morphological features such as a large multinucleated nucleus and abundant cytoplasm.

Certain biomarkers, including CD30 and CD15, are typically expressed on the surface of Reed–Sternberg cells. These markers are also used to detect Hodgkin disease [40]. It should be noted that the biomarkers for Reed cells can vary, and not all cells express identical markers [41,42].

The presence of Reed–Sternberg cells in the lymphatic microenvironment is associated with an inflammatory response characterized by the infiltration of immune cells and a variety of molecules that regulate different cell signaling processes [43,44].

B-lymphocytes, known for their classic characteristics, play a pivotal role in the humoral immune response by producing antibodies that aid various immune pathways. These cells exhibited uniform morphology, characterized by a relatively large nucleus and minimal cytoplasm. Unlike Reed–Sternberg cells, B lymphocytes do not have a multinucleated or lacunar appearance. In addition to morphological differences, B lymphocytes express specific surface markers including CD19, CD20, and CD21. These markers were used to identify and differentiate the cell population [45,46].

Thus, Reed–Sternberg cells and B lymphocytes exhibit unique characteristics in terms of their origin, morphology, function, surface markers, and association with various health and disease conditions [47].

Reed–Sternberg cells constitute only a minor portion of the tumor tissue and are a characteristic feature of HL. The tumor milieu also encompasses other cell types, including inflammatory and normal cells. These cells engage in a complex signaling system [48,49,50].

### 3.1. Molecular Networks in HL

In the previous section, we discussed the complex signaling pathway of HL, which involves cells and biomolecules that directly orchestrate tumor growth [51]. Not only does this includes Reed–Sternberg cells, a lineage derived from B lymphocytes that exhibit genetic alterations and abnormal protein expression, as well as other cells found within the tumor microenvironment. These cells include T helper lymphocytes, dendritic cells, macrophages, and eosinophils, which are among the various immune cells, tumor cells, stromal cells, and chemical mediators, including cytokines, chemokines, histamines, prostaglandins, leukotrienes, and growth factors. Histopathological analyses have revealed that the composition of these cell populations varies depending on the specific characteristics of each classical HL subtype [52,53].

The tumor microenvironment in nodular sclerotic HL is characterized by fibroblast-like cells and fibrosis. In contrast, the tumor microenvironment of mixed-cellularity HL comprises polymorphous reactive infiltrates, including B cells, T cells, neutrophils, histiocytes, plasma and mast cells. In lymphocyte-depleted HL, the tumor microenvironment typically comprises histiocytes, irregular fibrosis, and high concentrations of follicular dendritic cells. The tumor microenvironment of lymphocyte-rich HL is variable but usually contains histiocytes and lymphocytes [54,55].

HL comprise a diverse cell population, including characteristic lymphoid and myeloid lineage cells. These tumor cells can engage with the vascular microenvironment, thereby influencing angiogenesis, immune responses, and inflammation associated with HL [56].

Despite their reduced abundance in HL compared to other lymphocyte types, normal T and B lymphocytes can still be detected in the tumor environment. The interaction between Reed–Sternberg cells and the surrounding lymphocytes significantly influences the progression and clinical behavior of HL [53,57].

T lymphocytes present at the tumor inflammatory infiltration site significantly regulate the immune response by promoting and inhibiting the proliferation of Reed–Sternberg cells. T lymphocytes can be categorized into various subpopulations, including helper T lymphocytes (CD4+) and cytotoxic T lymphocytes (CD8+), each of which plays a unique role within the HL microenvironment. CD4+ cells secrete cytokines that influence the growth and survival of Reed–Sternberg cells, whereas, CD8+ can eradicate tumor cells [58,59].

Cytokines frequently associated with tumor inflammation include interleukin (IL)-13, IL-10, transforming growth factor (TGF)-β, and IL-5. IL-13, a cytokine secreted by type 2 helper T (Th2) lymphocytes, is associated with Th2-mediated immune responses [60]. In HL, IL-13 can stimulate adhesion molecule expression in Reed–Sternberg cells and induce the production of other cytokines, such as IL-10, potentially contributing to Reed–Sternberg cell survival [46,60]. IL-10, an anti-inflammatory cytokine, is primarily secreted by CD4+, particularly the Th2 subpopulation. It can modulate the immune response by inhibiting the activation of immune cells such as CD8+, thereby fostering an environment conducive to Reed–Sternberg cell survival [61]. TGF-β, a multifunctional cytokine, is produced by various cells, including CD4+ cells. In HL, TGF-β may promote Reed–Sternberg cell survival by inhibiting apoptosis (programmed cell death) and stimulating the production of growth factors that encourage cell proliferation [62,63]. IL-5, although less frequently discussed in relation to other cytokines, is primarily secreted by Th2 cells and is associated with allergic and inflammatory immune responses. In HL, IL-5 may contribute to Reed–Sternberg cell survival by influencing the inflammatory microenvironment [64,65].

The HL microenvironment is notably complex, with the release of cytokines by CD4+ cells constituting only one aspect of this intricate scenario. The interplay between Reed and Sternberg cells and various immune cells, coupled with the release of cytokines and other mediators, plays a significant role in disease progression. Understanding these interactions is crucial for developing targeted therapies and enhancing HL treatment strategies [52].

HL is characterized by a disruption in the microenvironment, where Reed–Sternberg cells contribute to the release of cytokines such as IL-13, IL-10, and TGF-β, along with chemokines and growth factors. These elements alter the cellular composition and immune functions [56,64]. Reed–Sternberg cells produce and release chemokines that attract other immune cells to the tumor site, which in turn negatively affects the recognition and attack of tumor cells [52]. In addition to the aforementioned cytokines, Reed–Sternberg cells secrete survival factors, such as B-cell-derived growth factor and vascular endothelial growth factor (VEGF). These factors promote the survival and proliferation of tumor cells, aiding their nutritional and oxygen supply [66,67].

Alterations in the microvasculature associated with endothelial factors have been observed in patients with HL. These changes correlate with an increase in vascular density and the expression of endothelial molecules [68]. Studies have explored the expression of endothelial factors including VEGF and platelet-derived growth factor (PDGF). These factors contribute to angiogenesis and may participate in the creation of new blood vessels within the tumor environment [68].

Platelet and coagulation cascade proteins are associated with a procoagulant state in HL, which is characterized by heightened platelet activation and increased expression of coagulation cascade proteins. This alteration in blood hemostasis could lead to a hypercoagulable environment within the tumor, potentially promoting the survival and progression of lymphoma [69].

The fibrinolytic pathway, which is responsible for the degradation of blood clots, has been observed to be activated in HL. This activation may be linked to the increased expression of fibrinolytic proteins, such as tissue plasminogen activator. Such an increase could potentially be associated with vascular remodeling and tumor invasion, thereby contributing to tumor progression [70].

Numerous signaling pathways have been examined in HL, including those implicated in cell proliferation, survival, immune responses, and interactions with the microenvironment. These pathways could potentially influence vascular biology and angiogenesis in HL [71,72].

Reed–Sternberg cells exhibit abnormal proliferation, which contributes to HL progression. The JAK-STAT pathway (Janus kinase signal transducer and activator of the transcription) plays a significant role in this process. Cytokines such as IL-13 and IL-10 bind to their respective receptors, thereby activating JAK. This activation leads to STAT phosphorylation. Once phosphorylated, STAT migrates to the nucleus where it regulates the expression of genes associated with cell proliferation [73].

The extended survival of Reed–Sternberg cells is a characteristic of HL. The NF-κB pathway, stimulated by chemokines and cytokines discharged by Reed–Sternberg cells and their microenvironment, facilitates the expression of genes associated with survival and resistance to apoptosis, including Bcl-2 [52,74].

Therefore, the PD-1/PD-L1 pathway is important. Reed–Sternberg cells express PD-L1, which upon binding to PD-1 on T lymphocytes, inhibits their activation, thereby diminishing the anti-tumor immune response [75].

The hypoxia-inducible factor-1α pathway is triggered in the tumor microenvironment when oxygen supply is low. This activation induces the expression of genes that facilitate cellular adaptation to hypoxic conditions. Among these, angiogenic growth factors encourage the development of new blood vessels to nourish the tumor [76].

TGF-β also significantly contributes to the interaction of Reed–Sternberg cells with the microenvironment. This factor enhances the production of extracellular matrix components, potentially facilitating the creation of a fibrous environment that safeguards tumor cells [77].

Although no study has comprehensively addressed these factors, numerous studies have explored these aspects individually, offering valuable insights into the correlation between vascular biology and HL [65,68,78,79].

Understanding the interactions between various cells and their microenvironments in HL is vital to the development of targeted therapies. Certain therapeutic strategies aim to adjust the body’s immune response to enhance its efficacy in eliminating the Reed–Sternberg cells. This could entail the administration of specific monoclonal antibodies, cytokine inhibitors, or other immunomodulatory agents [80].

### 3.2. Vascular Biology of HL and Survival Rates

Intricate molecular configurations evident in the structural and functional networks of vascular biology have been the focus of numerous studies in various fields. The heightened interest in phenomena observed in vascular biology can be partly attributed to the potential to execute more decisive interventions for the treatment and prevention of various diseases, including cancer [81]. This study discusses the correlation between the survival rate of patients with HL and the functional relationships between certain proteins in vascular biology (Table 1).

The structures discussed included tumor necrosis factor (TNF) receptor superfamily members 8 (CD30), 13 B (CD267), 17, 11A, IL-13, IL-13 receptor subunit α1, IL-21, IL-21 receptor, and tissue factor TF/TNF/IL-1/E-selectin. Other structures of interest are the effector cell protease receptor-1 (EPR-1), lymphotoxin-α, CXCL10, CXCL12, CCL21, ELAM-1/VCAM-1, and neutrophil release extracellular traps. Additionally, PAR-2, programmed cell death 1 ligands 1 and 2, latent membrane protein 1, BDNF/NT-3 growth factor receptor, PDGF-α, and MHC class II transactivators were included (Table 1).

In the field of vascular biology, certain protein structures that influence the tumor niche are particularly noteworthy. These include those involved in inflammatory activities such as recruitment, adhesion, signal transduction, immune response suppression, chemokine modulation, cytokine modulation, permeability, and MHC-II regulation. Additionally, proteins that play a role in angiogenesis and metastasis, such as CXCR4, are important. The control of intracellular calcium influx and DNA synthesis by CXCL10 also merits attention, given its crucial role in cell proliferation and invasion [82].

CD30, within the realm of vascular biology, is linked to inflammatory processes and endothelial functions. It is notably overexpressed in Reed–Sternberg cells, which is a characteristic feature of HL. This overexpression has significant therapeutic implications given the existence of a monoclonal antibody that specifically targets CD30 and is utilized in the treatment of HL [83].

CD267, referred to as 13 B or CD269, belongs to the TNF receptor family. Although its precise connection to vascular biology remains unclear, it is instrumental in modulating immune responses, particularly in regulating B-cell maturation. In relation to HL, CD267 could potentially influence the interaction between tumor cells and immune system cells [84].

The NaïveL-13 receptor α1 subunit (IL-13Rα1) is a crucial cell surface protein that binds to IL-13, playing a pivotal role in cellular signal transmission. Within the realm of vascular biology, the expression of IL-13Rα1 in endothelial cells could be significant in mediating the effects of IL-13 on vascular permeability, angiogenesis, and the inflammatory response. In the case of HL, IL-13 has been linked to the tumor microenvironment and interplay between tumor cells and immune system cells [85]. Some evidence suggests that Reed–Sternberg cells secrete IL-13, which could have implications for altering the tumor microenvironment, impacting the immune response, and influencing tumor cell survival [86].

IL-21, another cytokine produced by helper T cells, plays a crucial role in regulating immune responses and facilitating communication among immune cells. Although IL-21 does not have a direct connection to vascular biology, it can affect immune responses that subsequently influence vascular processes, including inflammation. The IL-21 receptor, primarily expressed on immune cells, such as T lymphocytes and B cells, may not play a direct role in vascular biology [87]. However, IL-21 activity can indirectly affect the immune response, which has implications for vascular regulation [88].

Tissue factor (TF), TNF-, and IL-1 are cytokines that play crucial roles in inflammatory responses and vascular regulation [89]. These cytokines are released in response to tissue damage or inflammation, which results in immune and vascular responses. These responses include endothelial cell activation, vasodilation and increased vascular permeability [90,91]. E-selectin, an adhesion molecule expressed on endothelial cell surfaces, is instrumental in the interaction between inflammatory cells and endothelium during an inflammatory response. It facilitates the adherence and migration of immune cells to the sites of inflammation, thereby contributing to the regulation of immune and vascular responses [92]. In HL, cytokines such as IL-21, TNF, and IL-1 may be implicated in the inflammatory responses of the tumor microenvironment. This involvement may, in turn, influence the interaction between Reed–Sternberg cells and the immune system. The expression and release of these cytokines can affect the progression of HL and the response to treatment [93].

EPR-1, a protein derivative, serves as an activation marker of Reed–Sternberg cells. Concurrently, it is associated with the upregulated expression of the leukocyte cell surface receptor for coagulation protease Factor Xa. This study also discussed lymphotoxin-α and vascular biology [94].

Lymphotoxin-α, produced by activated T cells, plays a role in modulating the immune response. Despite their lack of direct involvement in vascular biology, the immune responses they trigger can influence this field. In the context of inflammation, lymphotoxin-α may facilitate the activation of endothelial cells, which constitute the lining of blood vessels. This process affects vascular permeability, immune cell adhesion, and cell migration. In the case of HL, lymphotoxin-α may facilitate communication between tumor cells, such as Reed–Sternberg cells, and immune system cells within the tumor microenvironment. The expression and release of α-lymphotoxin can modulate the inflammatory response in HL, potentially impacting the interaction between tumor cells and the adjacent immune system [95,96].

Chemoattractant molecules, such as CXCL10, CXCL12, and CCL21, which are known for their role in attracting neutrophils or monocytes, also serve distinct functions in HL and vascular biology. CXCL10 regulates the migration of immune cells, including activated T lymphocytes, in response to inflammation. This chemokine potentially influences the chemotaxis of inflammatory cells towards injury or infection sites, thereby altering vascular biology in inflammatory settings. In the context of HL, CXCL10 expression has been detected in Reed–Sternberg cells and in cells within the tumor microenvironment. This cytokine may affect the immune response and the interactions between tumor cells and the adjacent immune system in HL [97,98].

CXCL12 significantly influences the migration and retention of hematopoietic stem cells within the bone marrow and plays a role in angiogenesis and blood vessel formation. Its expression in the tumor microenvironment of HL may contribute to the development of this environment and affect the migration of both immune and tumor cells within the affected tissue. Another chemokine, CCL21, is implicated in the migration of immune cells to lymphoid and inflammatory tissues, including the lymph nodes. It is instrumental in the organization of germinal centers within the lymphoid organs. The expression of CCL21 within the HL microenvironment may affect the organization of tumor and immune cells within the affected lymph nodes, potentially influencing disease progression and specific immune responses [66,99]. The other molecules associated with HL and vascular biology are listed in Table 1. The potential effects of biomolecules associated with vascular biology and classic HL can be seen in Figure 2.

**Table 1 cancers-15-05299-t001:** Biomolecular dysregulation to ensure survival in Hodgkin’s lymphoma and the potential association of biomolecules with vascular biology.

Study	Proteins	Tumor Activity	Vascular Biology
[55,100,101]	TNF receptor superfamily member 8 (CD30), 13B (CD267), 17 and 11A	Induces paracrine activation of NF-κB, thereby supporting HRS cell survival.	CD30 and CD30L were involved in pulmonary vascular remodeling and inflammatory response in COPD.
[55,102,103]	IL-13, IL-13 receptor subunit α1	Autocrine and paracrine activation of JAK–STAT pathways support enhanced cell growth.	Chemokine modulation in vascular smooth muscle cells; selectively induces vascular cell adhesion molecule-1 expression.
[55]	IL-21, IL-21 receptor	Autocrine and paracrine activation of JAK–STAT pathways support enhanced cell growth.	
[14]	Fator tecidual–TF/TNF/IL-1/E-selectina	Coagulation disorders.	Endothelial modulation, permeability, and chemotaxis.
[94,104]	Effector cell protease receptor-1	Activation marker for Reed–Sternberg cells.	Leukocyte cell surface receptor for the coagulation protease Factor Xa.
[105]	lymphotoxin-α	HRS cells secrete lymphotoxin-α which acts on endothelial cells to upregulate the expression of adhesion molecules that are important for T-cell recruitment.	Regulates lymphocyte growth and function.
[106]	CXCL10, CXCL12, CCL21	Were expressed on the malignant cells and/or vascular endothelium.	(CXCL10) Inducing intracellular calcium influx, DNA synthesis, cell proliferation, and chemotaxis; (CXCL12) cell migration and its binding to the CXCR4 receptor is involved in the process of tumor growth, angiogenesis, and metastasis; (CCL21) chemoattractant to guide naïve CCR7 expressing T cells to the T-cell zone.
[14]	ELAM-1/VCAM-1	Pronounced in cases of nodular sclerosis and was associated with a significantly higher content of perivascular neutrophils.	ELAM-1 plays an important role in recruiting leukocytes to the site of injury. (VCAM-1) Mediates adhesion of lymphocytes, monocytes, eosinophils, and basophils to vascular endothelium. It also acts on leukocyte-endothelial cell signal transduction.
[13]	Neutrophil release extracellular traps (NETs); PAR-2	NETs associated with the inflammatory process.	(NETs) Innate immunity pathogen containment mechanism; (PAR2) fibroblast proliferation, regulation of cytokines and vasodilation.
[55,107]	Programmed cell death 1, ligand 1 and 2	Genomic amplification and rearrangements.	Suppression of the immune response.
[108,109]	Latent membrane protein 1	Activation of the NF-κB pathway, thereby supporting survival of HRS cells in EBV-positive c’L.	It is not a human protein, it is a viral protein found in the Epstein–Barr virus.
[55]	BDNF/NT-3 growth factor receptor	Involved in cell adhesion, proliferation, and extracellular matrix remodeling.	
[110,111]	Platelet-derived growth factor receptor-α	Involved in cell adhesion, proliferation, and extracellular matrix remodeling.	Cell growth, proliferation, differentiation, and migration. Maintenance of various tissues in the body, including the nervous system, kidneys, and blood vessels.
[101,112,113]	MHC class II transactivator	Loss of effective antigen presentation.	Transcriptional coactivator that plays a critical role in the adaptive immune system by regulating the expression of MHC class II molecules.

### 3.3. Preclinical Models of HL

Various preclinical models have been employed to investigate the pathophysiological pathways involved in HL and identify potential therapeutic targets. These studies included a diverse range of cell profiles, computational models, and experimental animals. The animals used in these studies included mice, rats, and non-human primates, with mice being the most frequently model used to date.

Numerous mouse models have been used to investigate HL and to gain a deeper understanding of its pathogenesis, molecular attributes, and treatment responses. NOD/SCID mice were used as the HL model. These mice exhibited immune system deficiencies, rendering them ideal for studying engrafted human cells [114].

Human HL can be grafted onto specific animals to study tumor growth and therapeutic responses. NOD/SCID/gamma (NSG) mice are variants of the previously discussed lineage. These mice, a variant of NOD/SCID, exhibit more pronounced immune deficiency, including the absence of T lymphocytes, B cells, and natural killer cells. They are used to investigate human cell xenografts and their interactions with the tumor microenvironment [115,116,117].

Athymic nude mice are frequently used because of their low immunological resistance. In 2019, a research group explored the role of CCL5-CCR5 signaling in the interactions between monocytes, MSCs, and classical Hodgkin lymphoma (cHL) cells. This study employed cell lines in culture, female athymic nude/nude mice, and male NOD/SCID gamma chain-deficient (NSG) mice. The study concluded that maraviroc, when used to block the CCL5-CCR5 pathway, effectively reduced tumor mass [118]. In a separate study involving male NSG mice, researchers sought to determine the efficacy of trabectedin on HL xenografts. They concluded that pharmacological agents inhibited tumor growth by 75% and reduces angiogenesis [119].

Transgenic study models, created by introducing specific genes associated with HL into mice, facilitate the investigation of signaling pathways implicated in pathogenesis. Additionally, these models enable examination of the impact of these genetic alterations on lymphoma development.

Knockout models created by deleting a specific gene serve as valuable tools for understanding the role of individual genes in the pathogenesis of diseases, such as HL. In 2022, the authors assessed the correlation between BCOR loss-of-function mutations and classical HL. Using methods such as AID-Cre mice, the study concluded that BCOR loss may play a role in cHL pathogenesis and GC-B cell homeostasis [120].

In 2020, Sakai et al. [121] conducted a study utilizing lyn+/−, lyn−/−, and Ig-ganpTg mice with a C57BL/6 background. The objective of this study was to ascertain the potential correlation between lymphoid tumor development and the emergence of biphenotypic characteristics of B cells and macrophages analogous to human LH. The authors concluded that Ig-gan pTg mice represent a novel cytological model for investigating the cytopathological etiology and oncogenesis of HL.

In the reviewed studies, tumor xenograft experiments were frequently employed, particularly in immunocompromised animals [118,119]. Additionally, spontaneous development [121] or conditions that promote tumor establishment have been observed, such as the conditional loss of Bcor driven by AID-Cre in GC B cells [120].

In vitro models facilitate the evaluation of various standardized and laboratory-controlled cellular characteristics, including those of human cells, which are frequently used in HL studies. These models are instrumental for testing the efficacy of drugs and targeted therapies. However, because of the intricate network of control processes associated with HL and the potential cellular and molecular interactions within a complex biological system, it remains challenging to completely exclude the necessity for evaluation in experimental animals.

### 3.4. Emerging Therapies and Clinical Perspectives for HL

Given the findings from preclinical studies, especially the positive and significant temporal correlations, we now discuss the current targets in clinical investigations. The literature pertaining to clinical trials and research protocols was also reviewed. For clinical trials, the literature collection in the Medline/Pubmed database was consulted between the period 2016 and 2023, using the descriptors “Hodgkin Disease” and “Therapeutics”, as well as using their synonyms registered in “Mesh”. Nine studies were evaluated, five of which were selected for evaluation of clinical trials on HL. The predominant research phase was Phase II. Efforts have largely been aimed at improving interventions for classical relapsed/refractory HL.

Among these studies, 100% of the interventions demonstrated promise in advancing new clinical trials and/or other scientific evaluations. Therefore, we conducted a survey of the Cochrane database to verify the distribution of clinical records in recent years, using the same search strategy previously described for searching Medline/Pubmed. A total of 830 clinical trials will be conducted between January 2016 and October 2023. The drug and/or intervention, country, phase of the study, objective, and expected date for the end of the study were reported. Seventeen clinical protocols were described; 38.89% of the approaches were from groups in the United States, followed by China and Germany (22.22% each), and Italy, Canada, and Mexico (5.56% each). The largest number of trials were in phase II (52.63%), followed by phases III (31.58%), I (10.53%), and IV (5.26%). Descriptions of the objectives and interventions are summarized in Table 2.

Notably, research continues to advance and new therapies for HL are being developed with the aim of improving interventions. These approaches include immunotherapy, therapy with chimeric antigen receptors (CAR-T), and combined therapies. Immunotherapy is an innovative approach involving the use of medications that stimulate the patient’s immune system to fight cancers, including HL [122]. Immune checkpoint inhibitors, including pembrolizumab and nivolumab, have demonstrated efficacy in patients with refractory or relapsed HL [123,124]. Monoclonal antibodies, such as brentuximab vedotin, are effective in treating HL, and new monoclonal antibodies and targeted therapies are being developed to increase treatment options [125].

CAR-T cell therapy is a promising approach that involves modifying a patient’s T cells to attack cancer cells [126]. Clinical trials are currently underway to evaluate the efficacy of CAR-T cells in the treatment of HL. In a phase I clinical evaluation, the authors concluded that CAR-T cell therapy was safe, feasible, and effective in relapsed or refractory lymphoma, indicating the need to recruit patients for further clinical investigations [127].

In a second phase I clinical study, the authors corroborated the therapeutic value of CAR-T cells in HL. The clinical investigation included nine patients, seven of whom had progressed to the disease stage during intervention with brentuximab, and reported that CD30 CAR-Ts treatment is safe and can lead to favorable clinical responses in patients with HL [128]. A third clinical trial, currently in phase I/II, demonstrated that intervention with CAR-T cells had a high rate of durable responses with an excellent safety profile, highlighting the feasibility of extending CAR-T cell therapies to HL beyond canonical B cell neoplasms [129]. In this clinical evaluation, the authors aimed to evaluate CD30 CAR-T cell therapy as an intervention for HL. They included 41 patients who received CD30 CAR-Ts and had already undergone interventions, including brentuximab vedotin, checkpoint inhibitors, and autologous or allogeneic stem cell transplantation [129].

CAR-T therapies can be improved with greater specificity and sensitivity to tumor characteristics. An emerging and promising tool is the clustered regulatory interspaced short palindromic repeat/CRISPR-associated protein 9 (CRISPR/Cas9). This technology holds promise for advancing interventions for different types of cancers, including HL, owing to its flexibility, simplicity, high efficiency, and multiplexing capability in precise genome editing. CRISPR/Cas9 technology enables the construction of universal allogeneic CAR-T cells by disrupting inhibitory signaling to increase potency and facilitates the exploration of new, safer, and more controllable CAR-T cells [130].

Intervention protocols may vary according to the patient’s individual situation, and intervention with CAR-T cell therapy remains promising, especially in patients with relapsed and refractory classical HL. New multicenter evaluations may strengthen, become more consistent, and clarify the clinical variations in classic HL [131].

Neoantigen and messenger RNA (mRNA) therapies are promising approaches in the field of immunotherapy and have been studied for the treatment of several types of cancer, including HL [132]. However, it is important to highlight that these therapies are in the early stages of research and development, and several limitations and challenges remain to be overcome before they are widely used in the treatment of HL [133].

Tumor-associated macrophages (TAMs) are potential therapeutic targets [134,135,136] because of their critical roles in the lymph node microenvironment in HL and other types of cancer [137]. The relationship between TAMs and HL has been a topic of study and research with significant implications for the treatment of the disease [138]. These cells are recruited by the tumor and may play an important role in promoting lymphoma progression [134,139]. TAMs secrete pro-inflammatory factors and promote a tumor microenvironment favorable for the growth of malignant HL cells [135]. Given their role in HL progression, TAMs have become the focal point of targeted therapy [140].

Therapeutic strategies modulate the activity of TAMs in several ways, such as by inhibiting the signaling pathway responsible for recruiting TAMs to the tumor microenvironment, reducing the activation of TAMs or their polarization towards an anti-inflammatory state, and stimulating the activation of macrophages with antitumor properties [141,142].

A few therapies in the research phase or clinical trials have been designed to target TAMs for the treatment of HL. For instance, signaling inhibitors, such as colony stimulating factor 1 (CSF-1R) inhibitors, can block the activation and recruitment of TAMs, and immunotherapies, such as immune checkpoint therapy, can modulate the immune system response, consequently affecting the function of TAMs in the tumor microenvironment [57,134,143,144]

Treatment with high-dose liposomal doxorubicin is an alternative intervention that has shown benefits in classical HL with a high tumor burden. Liposomal doxorubicin is associated with rapid accumulation at high levels in TAMs of the lymphadenopathic and reticuloendothelial systems. Furthermore, initial therapy with increasing doses of liposomal doxorubicin is economical, because it reduces the number of patients who remain positron emission tomography scan-positive at the interim assessment, thereby reducing the need for subsequent aggressive treatments [145].

An advantage of liposomal doxorubicin is that it is designed to specifically target tumor cells, thereby reducing its toxicity to healthy cells [146]. This may create a more favorable environment for the immune system to attack the tumor, including modifying the activity of TAMs [145].

Another alternative is neoantigen therapy, which involves identifying specific genetic mutations in a patient’s tumor and creating a vaccine or targeted therapy to target these mutations, thereby stimulating the immune system to attack cancer cells [147]. However, not all tumors have easily identifiable mutations, making it difficult to select neoantigens for targeting. Neoantigen therapy is highly personalized for each patient, making the process expensive and time-consuming. The response to neoantigen therapy may vary from patient to patient and not all patients benefit equally [148].

In a recent phase I clinical trial, researchers evaluated the dose escalation of therapy with neoantigen-activated haploidentical T cells for the treatment of relapsed or refractory peripheral T-cell lymphoma and reported that the intervention was safe and effective; however, they addressed the need for new randomized multicenter evaluations, as well as the importance of describing the mechanisms of action in detail [149].

Another promising method is mRNA therapy, which involves delivering modified mRNA molecules to the patient to instruct body cells to produce therapeutic proteins such as cancer antigens that can trigger a tumor-directed immune response [150]. A major challenge of these methods is allowing mRNA to reach target cells and effectively translate them into therapeutic proteins. Furthermore, the immune system can identify mRNA as threats that trigger an unwanted immune response. Controlling the production of therapeutic proteins that are specific and effective in combating cancer is complex and difficult to standardize [151,152]. In the context of HL, these therapies may offer new treatment options, particularly for refractory or relapsed cases. However, studies are ongoing to better understand the safe and effective applications of these therapies for this specific disease.

**Table 2 cancers-15-05299-t002:** Perspectives for new therapies for HL based on clinical studies and protocols.

Study	Phase	Drug/Intervention	Country	Objective	Conclusions
[153]	II	Pembrolizumab added to chemotherapy with ifosfamide, carboplatin, and etoposide	USA	Assessment of complete response rate using 18F-fluorodeoxyglucose positron emission tomography with computed tomography after salvage therapy for patients with relapsed or refractory classic HL.	The addition of pembrolizumab to chemotherapy with ifosfamide, carboplatin, and etoposide was well tolerated and highly effective compared to previous reports of chemotherapy only regimens, supporting further investigation in patients with relapsed or refractory classical Hodgkin’s lymphoma eligible for an autologous cell–cell transplant. stem.
[154]	II	Brentuximab vedotin	China	To evaluate the efficacy, safety, and pharmacokinetics of single-agent brentuximab vedotin in Chinese patients with relapsed/refractory classical HL or systemic anaplastic large cell lymphoma.	Brentuximab vedotin had a positive benefit–risk profile for Chinese patients, confirming it as a potential treatment option.
[155]	I/II	Nanoparticle albumin-bound (nab) paclitaxel	USA	To determine the safety and efficacy of nab-paclitaxel in patients with relapsed/refractory lymphoma, HL, and N-HL.	The study was terminated prematurely because the overall response rate was 10% with two partial responses. The maximum dose tested was well tolerated and grade 3/4 hematologic adverse events, including neutropenia 25%, thrombocytopenia 20%, and anemia 15%, were modest.
[156]	II	Combination of hyperbaric oxygen pretreatment with autologous peripheral blood stem cell transplantation	USA	To determine the safety of hyperbaric oxygen intervention and its effectiveness in reducing neutrophil and platelet-engraftment time compared to matched historical controls in patients with multiple myeloma, non-Hodgkin’s lymphoma, and Hodgkin’s disease eligible for auto hematopoietic cell transplantation.	Hyperbaric oxygen injection appears to be well tolerated in the context of high-dose therapy and auto hematopoietic cell transplantation. The authors also indicated that prospective studies are necessary to confirm the potential benefits of hyperbaric oxygen therapy in relation to earlier recovery of blood counts, reduction in mucositis and use of growth factors, and a cost–benefit analysis would be necessary.
[145]	II	Liposomal doxorubicin supercharge	Italy	To evaluate the impact of liposomal doxorubicin supercharge-containing therapy on interim fluorodeoxyglucose positron emission tomography responses in high-risk diffuse large B-cell lymphoma or classical HL.	Presents convincing evidence that up-front treatments with R-COMP and MBVD schedules, including increasing dosages of liposomal doxorubicin, are a “proof of concept” for testing them in large multicenter phase III clinical trials.
Responsible/expected completion date	Phase	Drug/intervention	Country	Objectives	Registry
González-Ramella, 2024 [157]	IV	Pentoxifylline	Mexico	To evaluate the potentiating effect of pentoxifylline on tumor apoptosis in combination with chemotherapy in pediatric patients and adolescents and young adults with HL.	NCT05490953
Sass et al., 2022 [158]	II	GHSG-AFM13	Germany	Demonstrate the efficacy of AFM13 with an optimized treatment schedule.	NCT02321592
Castellino et al./NCI (Nacional cancer institute), 2029 [159]	III	Brentuximabe vedotina	USA/Canada	To evaluate the use of brentuximab vedotin in chemotherapy combination for the treatment of HL in children.	NCT02166463
Zinzani et al./Merck Sharp & Dohme LLC, 2025 [160]	III	Pembrolizumbe (MK-3475)	USA	To evaluate pembrolizumab (MK-3475) in the treatment of participants with relapsed or refractory classical HL.	NCT02684292
Gillessen et al., 2022 [161]	II	Ruxolitinib	Germany	To evaluate the safety and efficacy of ruxolitinib in patients with recurrent classical HL.	NCT02164500
Herrera et al./City of Hope Medical Center, 2023 [162]	II	Nivolumab and brentuximab vedotin	USA	Evaluate the efficacy of nivolumab plus brentuximab vedotin consolidation after autologous stem cell transplantation in participants with relapsed/refractory HL.	NCT03057795
Ansell et al., 2026 [163]	III	Brentuximab vedotin/doxorubicin/ bleomycin/vinblastine/dacarbazine	USA	To evaluate two intervention protocols in the treatment of HL as first-choice therapy.	NCT01712490
Hochberg et al. New York Medical College, 2022 [164]	II	Brentuximab vedotin/ doxorubicin/vincristine/rituximab	USA	To evaluate the addition of brentuximab vedotin to combination chemotherapy in children, adolescents, and young adults with all stages of newly diagnosed HL.	NCT02398240
Prof. Dr. Andreas Garcia-Marquez et al. University of Cologne, 2023 [165]	II	anti-PDI-1 nivolumab	Germany	To demonstrate the efficacy of the two experimental treatment strategies and improve first-line treatment for early unfavorable cHL through the introduction of the anti-PD-1 antibody nivolumab.	NCT03004833
Cai, Q.; Sun, Yat-sen, 2026 [166]	II	Penpulimab and AVD/ sequential penpulimab and AVD	China	To evaluate the safety and effectiveness of penpulimab combined with AVD in patients with newly diagnosed advanced classical HL.	NCT05949931
Weidong, H., Chinese PLA General Hospital, 2026 [167]	I/II	SHR2554 + SHR1701/SHR-1701	China	Phase I dose escalation, evaluate the safety and feasibility of SHR1701 in patients with relapsed or refractory classical HL. Phase II, evaluate the antitumor effect of SHR1701 alone or in combination with SHR2554.	NCT05896046
Gloria, G. Biosciences Co., Ltd., 2025 [168]	III	GLS-010	China	To evaluate the efficacy of GLS-010 in participants with relapsed or refractory classical HL.	NCT05518318
Akeso, 2026 [169]	III	Penpulimab	China	To evaluate the efficacy of penpulimab compared to standard chemotherapy.	NCT05244642
Sharp, M., LLC, D, 2031 [170]	III	Favezelimabe/pembrolizumabe	USA	Compare the efficacy of this co-formulated favezelimab/pembrolizumab (MK-4280A) with chemotherapy.	NCT05508867
Linfomi, F.I (ETS), 2028 [171]	I/II	Atezolizumab	Italy	To evaluate the safety of atezolizumab combined with the BEGEV regimen.	NCT05300282
GmbH, A, 2027 [172]	II	AFM13 em combinação com AB-101	Germany	To evaluate the safety and efficacy of AFM13 in combination with AB-101 in subjects with classic HL.	NCT05883449
Institute, N.C (NCI), 2026 [173]	II	Mosunetuzumab	USA	To compare mosunetuzumab to usual care (rituximab) to improve survival in patients with nodular Hodgkin lymphoma.	NCT05886036

## 4. Conclusions and Future Directions

This approach strengthens our understanding of the relationship between molecules linked to vascular biology and HL. The imbalance in the immune system and clinical status of the disease, as well as its particularities for each HL type, are orchestrated by a complex network of molecules, many of which directly or indirectly participate in local or systemic functions of the vascular system.

We also highlight advances in scientific research using different methods and types of studies, including preclinical studies. Studies that seek models that guarantee more extensive evaluations in biological systems for different variables or biomarkers, associating dual effects for the molecules that integrate vascular biology and HL, can be a promising route for evaluations that can substantially collaborate in response to the prognosis of this cancer type, even in new intervention perspectives for this disease.

## Figures and Tables

**Figure 1 cancers-15-05299-f001:**
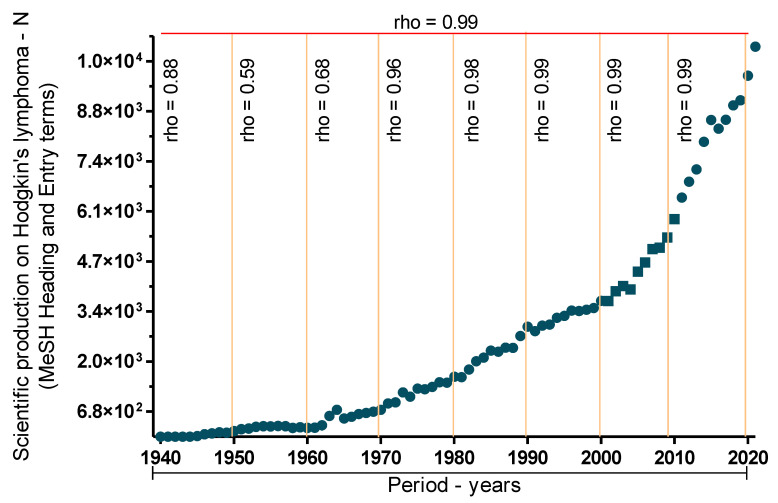
Evolution towards the absolute frequency of scientific approaches to Hodgkin’s lymphoma. Data on the number of publications from 1940 to 2021 that mention, or address Hodgkin’s lymphoma were obtained from the Medline/Pubmed database. Data were stratified into eight blocks containing a decade for each block. All distributions were correlated by Spearman’s test. The significance level used was 5%.

**Figure 2 cancers-15-05299-f002:**
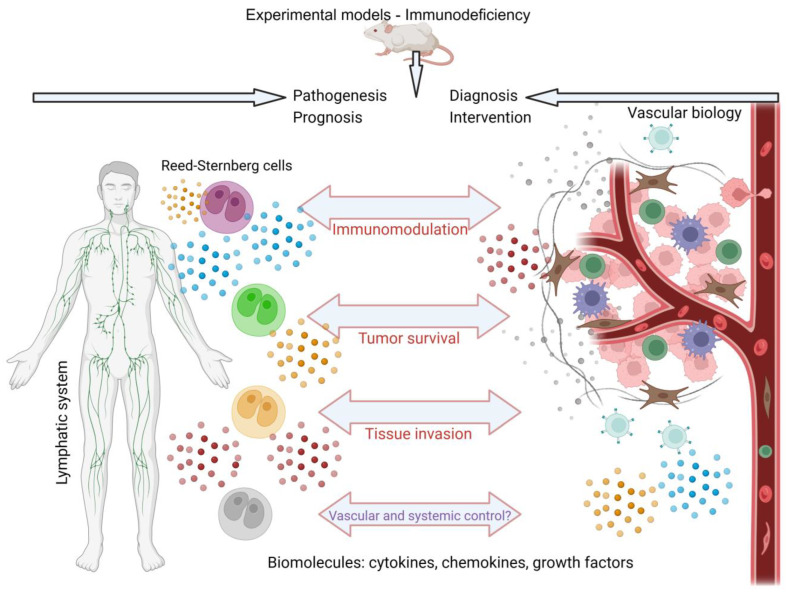
Schematic model to demonstrate the potential interaction between molecules that participate in tumor activity, present in classic HL, and vascular biology. Among the biomolecules found, cytokines, chemokines and growth factors stand out. The animal models for study most frequently use immunodeficient mice, where the pathogenesis, prognosis, diagnosis, and intervention for classic HL are evaluated. Created with BioRender.com.

## Data Availability

Data are available from a publicly accessible repository. The data presented in this study are available at https://osf.io/4r8n9/ (accessed on 3 July 2023).

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
