# Peer review of "Potential Associations between Vascular Biology and Hodgkin’s Lymphoma: An Overview"

_cancers, 2023, doi:10.3390/cancers15215299_

Round 1
Reviewer 1 Report
Comments and Suggestions for Authors
The review entitled ' Cellular and Molecular Associations Between Vascular Biology and Hodgkin's Lymphoma: Signaling Pathways and Disease Study Models’ by Rodrigues et al is very interesting and gives an overview of the Hodgkin's Lymphoma and the recent developments. However, The authors could improve the review article.
1. The article focusses on the vascular biology aspects and the signaling mechanisms involved in HL. A schematic picture describing the various cytokines, chemokines regulated in the paradigm would be more illustrative.
2. The introduction of HL is very nicely outlined but needs to outline the key drivers that cause HL. The authors mentioned only about the Reed-Sternberg cells, but what are the genetic aberrations or mutations in these cells need to be described.
3. The authors need to add a separate section for emerging therapies and outline some of them such as CAR-T therapy, A schematic representation would be more impactful.
4. The recent advances of CRISPR-Cas9 can be a potential strategy to Knock out or knock in desired target antigens in HL. A brief overview of the available therapeutic interventions is required.
5. There are new modalities such as neo antigen therapy or mRNA therapy, the authors could comment and discuss if they can be used and their limitations.
6. The review can describe about the ongoing clinical trials. A brief table explaining the targets and the outcome of the study would be a great addition to the review article.
7. There are some minor English spelling errors in the article such as Protein spelling in Table 1, Activation of the NF-kB pathway.
Comments on the Quality of English LanguageThere are some minor English spelling errors in the article such as Protein spelling in Table 1, Activation of the NF-kB pathway.
Author Response
Thank you very much for taking the time to review this manuscript and for providing pertinent comments and suggestions that allowed us to improve the manuscript. Please find our detailed responses below and the corresponding revisions and corrections highlighted in yellow in the re-submitted files.
- The article focusses on the vascular biology aspects and the signaling mechanisms involved in HL. A schematic picture describing the various cytokines, chemokines regulated in the paradigm would be more illustrative.
Answer: Thank you for the suggestion. The new version now includes a schematic model to represent and simplify the relationship between cytokines, chemokines, vascular biology, and HL. We remain open to any other suggestions.
- The introduction of HL is very nicely outlined but needs to outline the key drivers that cause HL. The authors mentioned only about the Reed-Sternberg cells, but what are the genetic aberrations or mutations in these cells need to be described.
Answer: Thank you for the comments. The introduction was restructured and now includes the requested items. The changes are highlighted in yellow.
- The authors need to add a separate section for emerging therapies and outline some of them such as CAR-T therapy, A schematic representation would be more impactful.
Answer: We appreciate your suggestion and have inserted another subsection (3.4) that addresses the list of clinical studies and perspectives for emerging interventions for HL.
- The recent advances of CRISPR-Cas9 can be a potential strategy to Knock out or knock in desired target antigens in HL. A brief overview of the available therapeutic interventions is required.
Answer: We appreciate the suggestion. We have included an important association of the tool (CRISPR-Cas9) with the improvement of technologies applied to HL intervention.
- There are new modalities such as neo antigen therapy or mRNA therapy, the authors could comment and discuss if they can be used and their limitations.
Answer: Thank you for the suggestion. We have included in the new section an approach to the potential and limitations for the use of mRNA and neo antigen therapies.
- The review can describe about the ongoing clinical trials. A brief table explaining the targets and the outcome of the study would be a great addition to the review article.
Answer: Thank you for the suggestion. We have followed your suggestion and added a new Table (Table 2) with the description for some clinical trials, allowing a broad vision for future approaches associated with interventions for HL.
- There are some minor English spelling errors in the article such as Protein spelling in Table 1, Activation of the NF-kB pathway.
Answer: Thank you for the recommendation. The new version has been thoroughly revised to improve the quality of language and spelling.
Reviewer 2 Report
Comments and Suggestions for Authors
The authors present a review on a potential interesting topic, however the review is not informative, repeats for the most part general notions on Hodgkin lymphoma, and when it finally comes to vascular biology just lists molecules expressed in Hodgkin lymphoma. It is not clear what drives the selection of the molecules included, as most play no role in vascular biology.
Author Response
Thank you very much for taking the time to review this manuscript. Please find detailed responses below and the corresponding revisions and corrections highlighted in yellow.
- The authors present a review on a potential interesting topic, however the review is not informative, repeats for the most part general notions on Hodgkin lymphoma, and when it finally comes to vascular biology just lists molecules expressed in Hodgkin lymphoma. It is not clear what drives the selection of the molecules included, as most play no role in vascular biology.
Answer: Thank you for your comment, it helped us improve the work presented. In the new version, in light of your comment, we have added a detailed explanation on the role of the molecules selected in the relationship between vascular biology and HL.
Reviewer 3 Report
Comments and Suggestions for Authors
Line 19: Objective of the study "is", rather than was. Line 59: survival of HL "patients". Figure 1: Scientific Production: "not clear what is meant by "Scientific production" Does the author mean "Scientific Publications"; needs some clarification. The figure is fine otherwise. Line 160: These cells? "are the authors referring to RS cells, or, B cells, or both (clarify). Line 179: "Chemical mediators" is vague, "perhaps give some examples", or clarify."
Author Response
Thank you very much for taking the time to review this manuscript. Please find detailed responses below and the corresponding revisions and corrections highlighted in yellow.
- • Line 19: Objective of the study "is", rather than was. Line 59: survival of HL "patients". Figure 1: Scientific Production: "not clear what is meant by "Scientific production" Does the author mean "Scientific Publications"; needs some clarification. The figure is fine otherwise. Line 160: These cells? "are the authors referring to RS cells, or, B cells, or both (clarify). Line 179: "Chemical mediators" is vague, "perhaps give some examples", or clarify."
Answer: Thank you for all the observations and suggestions. All recommendations were carefully followed (they are marked in yellow in the new version presented).
Reviewer 4 Report
Comments and Suggestions for Authors
The authors have presented the cellular and molecular connections between vascular biology and Hodgin's lymphoma in a very beautiful and detailed way.
The only objection concerns the abbreviations for cytokines and growth factors that are given in the abstract without having been previously introduced.
Author Response
Thank you very much for taking the time to review this manuscript.
- • The authors have presented the cellular and molecular connections between vascular biology and Hodgin's lymphoma in a very beautiful and detailed way.
Answer: We are happy with your comments and hope that the new version, presented after the modifications suggested by other reviewers, remains acceptable to you.